# Effectiveness of COVID-19 Vaccines in the General Population of an Italian Region before and during the Omicron Wave

**DOI:** 10.3390/vaccines10050662

**Published:** 2022-04-22

**Authors:** Cecilia Acuti Martellucci, Maria Elena Flacco, Graziella Soldato, Giuseppe Di Martino, Roberto Carota, Antonio Caponetti, Lamberto Manzoli

**Affiliations:** 1Department of Environmental and Prevention Sciences, University of Ferrara, 44121 Ferrara, Italy; cecilia.martellucci@unife.it (C.A.M.); mariaelena.flacco@unife.it (M.E.F.); 2Local Health Unit of Pescara, 65124 Pescara, Italy; graziella.soldato@ausl.pe.it (G.S.); giuseppe.dimartino@ausl.pe.it (G.D.M.); roberto.carota@ausl.pe.it (R.C.); antonio.caponetti@ausl.pe.it (A.C.); 3Department of Medical and Surgical Sciences, University of Bologna, 40100 Bologna, Italy

**Keywords:** SARS-CoV-2, COVID-19, vaccine, Omicron, cohort study, Italy

## Abstract

We performed a cohort analysis of the entire population of Abruzzo, Italy, to evaluate the real-world effectiveness of SARS-CoV-2 vaccines against infection, COVID-19 hospitalization or death, over time and during the Omicron wave. All resident or domiciled subjects were included, and official vaccination, COVID-19, demographic, hospital and co-pay exemption datasets were extracted up to 18 February 2022. Multivariable analyses were adjusted for age, gender, hypertension, diabetes, major cardio- and cerebrovascular events, COPD, kidney diseases, and cancer. During the follow-up (average 244 days), 252,365 subjects received three vaccine doses (of BNT162b2, ChAdOx1 nCoV-19, mRNA-1273 or JNJ-78436735), 684,860 two doses, 29,401 one dose, and 313,068 no dose. Overall, 13.4% of the individuals were infected with SARS-CoV-2 (*n* = 170,761); 1.1% of them had severe COVID-19, and 0.6% died. Compared with the unvaccinated, those receiving two or three vaccine doses showed an 80% to 90% lower risk of COVID-19 hospitalization or death. Protection decreased during the Omicron wave and six months after the last dose, but it remained substantial. Lethal disease was uncommon during the Omicron wave and in the young population, even among the unvaccinated. Some of the current policies may need a re-evaluation in light of these findings. The results from the Omicron wave will inevitably require confirmation.

## 1. Introduction

Vaccination is among the key preventive strategies against SARS-CoV-2 pandemic [1,2]. Phase 3 trials reported the efficacy of vaccines against symptomatic COVID-19 ranging from 67% to 95%, with acceptable safety profiles [3,4,5,6,7,8,9], and these findings were confirmed by several population studies [10,11,12,13].

However, several questions remain unanswered. Firstly, the length of follow-up is inevitably short, and vaccine long-term safety and effectiveness is uncertain. Indeed, a growing body of literature suggests that vaccine immunity may decrease over time as a result, among other factors, of the emergence and spread of new virus variants [14,15,16,17,18]. Secondly, it is unclear whether current vaccines are still effective against the virus variant B.1.1.259, commonly named “Omicron” [19], as early investigations found a decrease of 23% in the efficacy against Omicron-related hospitalizations [20,21]. Third, as a response to the potential waning of vaccine-induced immunity and Omicron spread, many countries introduced a third, booster dose of vaccine, the effectiveness of which remains to be evaluated in large population studies [22,23].

In this cohort analysis of the entire population of an Italian region, we evaluated the risk of infection, COVID-19 hospitalization and death of the unvaccinated subjects versus those who received different doses of SARS-CoV-2 vaccines, over time and during the Omicron wave.

## 2. Materials and Methods

This retrospective cohort study follows and expands a previous, preliminary analysis [11]. Here, we included all the subjects resident or domiciled in the Abruzzo region of Italy on 1 January 2020, with no positive SARS-CoV-2 swab before the start of follow-up, in order to evaluate the effectiveness of SARS-CoV-2 vaccination (one, two or three doses versus none) against infection (detected through RT-PCR-Reverse transcription polymerase chain reaction; tested through nasopharyngeal swabs by the accredited laboratories of the region), virologically confirmed COVID-19 severe disease, diagnosed by a specialist physician and requiring hospital admission, and COVID-19-related death (inside or outside the hospital) [24].

According to the Italian National immunization campaign plan, starting from 2 January 2021, Pfizer–Biontech BNT162b2, Oxford–AstraZeneca ChAdOx1 nCoV-19, Moderna mRNA-1273 and Janssen JNJ-78436735 vaccines were gradually administered to the population [25]. The Italian National Health System (NHS) provides vaccines administration, SARS-CoV-2 testing, and all healthcare services to its residents free of charge. Tests are mandatory for all individuals with suggestive symptoms such as fever or acute respiratory illness, for those that have been in contact with infected persons, and for all individuals returning from travel abroad. On 8 January 2022, a law decree established that, from February 15, vaccination was to become mandatory for all Italian citizens aged 50 years or more [26]. As the control group for these analyses was the group of unvaccinated persons, which may disappear as a consequence of this obligation, we extracted vaccine data before the mandate ruling (18 December 2021), and COVID data at the first available date after the mandate application: 18 February 2022.

From 2 January 2021 (date of the administration of the first vaccine dose), up to 18 December 2021:The subjects who received only one dose of BNT162b2, mRNA-1273 or ChAdOx1 nCoV-19 vaccines, were included in the group “1 dose”;The subjects who received only one dose of JNJ-78436735 vaccine, or only two doses of BNT162b2, mRNA-1273 or ChAdOx1 nCoV-19 vaccines, were included in the group “2 doses”;The subjects who received two doses of JNJ-78436735 vaccine, or three doses of BNT162b2, mRNA-1273 or ChAdOx1 nCoV-19 vaccines, were included in the group “3 doses”.

For the analyses evaluating the effectiveness of the various vaccines vs. no vaccines, the subjects who received two or three different vaccines were included in a global category named “mixed vaccines”.

The start of follow-up varied across vaccine categories, as all vaccine groups were separately compared versus unvaccinated subjects:For the analyses comparing unvaccinated versus those receiving only one vaccine dose (set of analyses “1 dose”), the start of follow-up was set 14 days after the single vaccine dose (to account for the time required for seroconversion) [27] for vaccinated subjects and 14 days after the first administration of the first vaccine dose (16 January 2021) for the group “unvaccinated”.For the analyses comparing unvaccinated versus those receiving two vaccine doses only (set of analyses “2 doses”), the start of follow-up was set 14 days after the second vaccine dose for vaccinated subjects and 14 days after the first administration of the second vaccine dose (31 January 2021) for the group “unvaccinated”.For the analyses comparing unvaccinated versus those receiving three vaccine doses (set of analyses “3 doses”), the start of follow-up was set 14 days after the third vaccine dose for vaccinated subjects and 14 days after the start of the mass administration of the third vaccine dose (17 September 2021) for the group “unvaccinated”.

For the outcome “infection”, the end of follow-up was set:−On 18 February 2022 for those who had no positive swabs during the follow-up;−On the day of the first positive swab for those who were not infected before the start of the follow-up (“uninfected” cohort).

For the outcome “COVID-19 hospitalization”, the end of follow-up was set on the day of the hospital admission, or on 18 February 2022, for all the subjects who were not hospitalized with COVID-19 related symptoms. For the outcome “death”, the end of follow-up was set on the day of the death with a positive swab, or on 18 February 2022 for all others. A schematic definition of the study groups, outcomes and follow-up is reported online in Appendix A.

To account for some of the main potential confounders of the association between vaccination and COVID-19 hospitalization or death [28], we used (a) the COVID-19 database, (b) co-pay exemption database and (c) administrative discharge abstracts from the last ten years to extract the following conditions for each resident: diabetes (ICD-9 cm codes in any diagnosis field—250.xx); hypertension (401.xx–405.xx); major cardiovascular or cerebrovascular diseases (410.xx–412.xx; 414.xx–415.xx; 428.xx or 433.xx–436.xx); chronic obstructive pulmonary diseases—COPD (491.xx–493.xx); kidney diseases (580.xx–589.xx); and cancer (140.xx–172.xx or 174.xx–208.xx).

### 2.1. Data Collection

All the information about vaccines, laboratory tests, demographic, anamnestic and clinical residents’ data are routinely entered into NHS official datasets, updated daily, and sent to the Italian Institute of Health [29]. We extracted all data from the official vaccination, COVID-19, demographic, hospital (Italian SDO) and co-pay exemption (Italian “Esenzioni Ticket” file) datasets of the Abruzzo region. Individual data were merged through encrypted fiscal code.

### 2.2. Statistical Analyses

The main analyses separately compared the risk of the primary infections, COVID-19 hospitalization and COVID-related death of unvaccinated subjects versus individuals who received one, two, or three vaccine doses. Each of these comparisons were stratified by age category (0–29y, 30–59y, 60 + y), vaccine type, and time since the last vaccine dose (≤ or >182 days, or six months, which was the duration of the “green pass” set by the Italian Government for subjects who recovered after a SARS-CoV-2 infection) [30]. The age ranges were identified to be consistent with the periodical reports of the Italian Institute of Health [31], and with the recommendations of the Italian Government [32], which identified subjects aged ≥ 60 years as priority targets for vaccination.

As the survival analyses could be biased due to the severe unbalance in the average follow-up duration between the unvaccinated individuals (367 days) and those who received two (181 days) or three (71 days) vaccine doses, and considering that more than 80% of the infections occurred during the restricted time span of the Omicron wave (the last 50 days of follow-up), we opted to use multiple logistic regression instead of the typical Cox proportional hazards analysis to evaluate the independent association between each outcome and vaccination [33]. All models were adjusted for age, gender, hypertension, diabetes, major cardio- and cerebrovascular events, COPD, kidney diseases, and cancer, all included a priori, and the results of such covariates were reported in the online Appendix A. Standard post-estimation tests were used to assess the validity of final models, performing multicollinearity and influential observation analyses (using standardized residuals, change in Pearson and deviance chi-square), and testing for potential interactions between vaccination and other covariates [34]. The goodness of fit was checked using Hosmer–Lemeshow test, and predictive power was assessed through C-statistics (area under the receiving operator curve).

The analyses were repeated, including only the events that occurred during the Omicron wave (from 27 December 2021 to 18 February 2022, when the proportion of Omicron variant in the available positive swabs was higher than 50%) [35]. In these analyses, all the subjects with a positive SARS-CoV-2 swab before 27 December 2021 were classified as “infected before the start of follow-up” and excluded.

A two-sided *p*-value of <0.05 was considered to be significant for all analyses, which were carried out using Stata, version 13.1 (Stata Corp., College Station, TX, USA, 2014).

## 3. Results

The STROBE flowchart of the study participants is shown in Figure 1. From 2 January 2021 to 18 December 2021, a total of 313,068 unvaccinated (24.5%) and 966,626 vaccinated (who received 2,156,216 doses) residents or domiciled in the Abruzzo region, Italy were included in the analyses. The main demographic characteristics of the population and the proportion of selected comorbidities are reported in Table 1. Overall, 29,401 subjects received only one vaccine dose (2.3% of the population), 684,860 received two doses (53.5%), and 252,365 received three doses (19.7%).

The majority of the population (63.9%) received only BNT162b2 doses; 13.0% received only mRNA-1273 doses; 11.5% ChAdOx1 nCoV-19, and 10.4% mixed vaccines.

During the follow-up, 13.4% of the population was infected with SARS-CoV-2 (*n* = 170,761): 19.1% of the unvaccinated, 13.8% of those receiving one dose, 13.1% of those receiving two doses, and 6.5% of those who received three doses (Table 2). During pre-Omicron waves (344 days of follow-up), we observed a total of 30,726 primary infections (89 per day), 1152 COVID-19 hospitalizations (3.3 per day; 3.8% of the infected), and 933 COVID-19-related deaths (2.7 per day; 3.0% of the infected). During the first 54 days of the Omicron wave, among the 1,245,097 subjects who were not infected before 27 December 2021, we observed 140,035 primary infections (2593 per day), 758 COVID-19 hospitalizations (14 per day; 0.5% of the infected), and 117 COVID-19-related deaths (2.0 per day; 0.1% of the infected).

The mean follow-up was 244 ± 99 days, but varied widely by vaccine group, being shortest (71 ± 18 days) for the analyses of the effectiveness of the third dose, which was widely administered from September, 2021.

### 3.1. Risk of Primary Infection

The highest incidence of infection was observed in the young population (0–29 years): 21.8%, as compared to 6.3% among the adults aged ≥ 60 years. In the multivariable analysis (Table 3), when adjusting for age, gender, and comorbidities, the risk of infection was significantly lower for vaccinated subjects. As compared to the unvaccinated, the individuals who received one, two, or three doses showed the following adjusted odds ratios (ORs), respectively: 0.71, 0.75, and 0.74, with 95% confidence intervals (Cis) ranging from 0.68 to 0.76. The results did not substantially vary by vaccine type or age category, but when the analyses were restricted to the Omicron wave, only the subjects who received three doses showed some degree of protection against infection (OR: 0.81; 95% CI: 0.80–0.83), while those who received two doses did not differ from the unvaccinated (OR: 0.98; 0.97–1.00). On the contrary, during the pre-Omicron waves, the risk of infection was significantly lower among the subjects who received any dose of vaccine (online Appendix A).

### 3.2. Risk of COVID-19 Hospitalization

During the follow-up, 1910 persons had a COVID-19 hospitalization (0.15% of the population; 1.1% of the infected subjects—Table 2). Among the 70,345 infected, young subjects, the incidence of COVID-19 hospitalization was lower than 0.1% (*n* = 24), rising to 6.2% among older adults. Before the Omicron wave, 5.3% of unvaccinated, infected subjects had severe disease (975/18,489); during the Omicron wave, this value decreased to 0.7% (295/41,281).

Overall, the frequency of COVID-19 hospitalization was considerably higher among the unvaccinated (0.41% of the population; 2.12% of the infected) as compared to those receiving two (0.06% and 0.44%, respectively) or three (0.06% and 1.00%, respectively) vaccine doses. Multivariable analyses confirmed a significantly and substantially lower risk of COVID-19 hospitalization for the subjects who received one (OR: 0.53), two (0.12) or three (0.21) vaccine doses, as compared with the unvaccinated (Table 3; all *p* < 0.001). The results were similar when the analyses were restricted to the infected persons, and stratified by vaccine type or age category. In contrast, during the Omicron wave (Table 3), and after six months from the last vaccine dose (Table 4), the effectiveness of two doses decreased to approximately 66% and 69%, respectively, although remained highly significant.

### 3.3. Risk of COVID-19-Related Death

During follow-up, 1050 persons with COVID-19 died (0.08% of the total population; 0.6% of the infected; 54.9% of the hospitalized subjects—Table 2). There were no deaths during follow-up in the population aged 0–29 years. Before the Omicron wave, 4.5% of the unvaccinated, infected subjects died (831/18,489); during the Omicron wave, this value decreased to 0.1% (41/41,281).

Overall, both mortality and lethality were considerably higher among the unvaccinated (0.28% and 1.46%, respectively) as compared to those receiving two (0.02% and 0.12%) or three (0.01% and 0.19%) vaccine doses. Multivariable analyses confirmed a significantly and substantially lower risk of death for the subjects who received one (OR: 0.49), two (0.06) or three (0.24) vaccine doses, as compared with the unvaccinated (Table 3; all *p* < 0.001). The results were similar when the analyses were restricted to the infected persons, and stratified by vaccine type or age category. A lower, yet considerable protection of two vaccine doses was observed six months after the last dose (OR: 0.25; 95% CI: 0.17–0.35; Table 4) and during the Omicron wave, where the number of deaths was too low to determine a lack of statistical power. Only the group receiving three doses showed a significantly lower risk of death compared to the unvaccinated (OR: 0.42; 95% CI: 0.26–0.68; Table 3).

### 3.4. Vaccine Safety

The regional surveillance system communicated that, between 2 January 2021 and 31 December 2022, the passive surveillance system received 179 spontaneous reports of possibly vaccine-related adverse events, 64 of which were defined as “severe”, with no deaths from 2,306,000 doses administered during the year 2021 (2.8 severe adverse event ×100,000 doses). The system did not provide any other data, including the proportion of severe adverse events whose correlation with vaccination was assessed and verified [36].

## 4. Discussion

This study evaluated the effectiveness of some of the most commonly used SARS-CoV-2 vaccines in the general population of an Italian region, followed for more than six months on average, in the context of a mass vaccination campaign. The main findings were: (a) compared with the unvaccinated, persons who received two or three vaccine doses showed a 80% to 90% lower risk of COVID-19 hospitalization or death; (b) during the Omicron wave, and six months after the last dose, protection against COVID-19 decreased but remained considerable; (c) the effectiveness against infection was modest at any dose; (d) the risk of a lethal disease was very small during the Omicron wave and in the young population, even among the unvaccinated; and (e) overall, the reported incidence of serious adverse events was very low.

Our results on the effectiveness of current vaccination schedule are consistent with the existing literature on pre-Omicron waves [18,21]. Since the first months of 2021, two doses of vaccines were able to confer a high real-world protection against severe or lethal COVID-19 in several population studies [10,11,12,13,37,38,39]. With regard to the increase in the effectiveness that may derive from a third dose, which has been extensively recommended worldwide, some preliminary reports from Israel, the UK, and USA have recently been published, indicating the very high effectiveness of booster doses against severe or fatal disease [23,40,41]. In this study, we observed a limited or null increase in the protection after a third dose, but the effectiveness was already very high after two doses, and the follow-up was relatively short. Concerning the potential decrease in protection caused by the Omicron variant, our findings were reassuring, in line with those from a few recent studies that also reported a high vaccine effectiveness against severe or lethal COVID-19 during Omicron predominance [23,40,41]. As both present and previous studies could only consider the first weeks of the Omicron wave, further data are inevitably required to confirm whether the current vaccines should still be considered as the pivotal strategy to control the pandemic, or whether new vaccines or policy changes are required.

The first reports of waning vaccine immunity emerged in October 2021, when Tartof et al. hypothesized that vaccine effectiveness against infection decreases with time from immunization, irrespective of the dominant virus variant [42]. In December 2021, another study reported no residual effectiveness against severe or lethal disease after more than seven months [16], and data from Israel suggested that waning immunity may also incur after booster doses [43]. In our study, we observed a reduction of 30% in effectiveness after six months from the last vaccine dose, but subjects who received two doses still showed a 70% lower risk of COVID-19 hospitalization compared with the unvaccinated, during both pre-Omicron and Omicron waves. Although the biological mechanisms of the waning of vaccine-induced immunity remain to be elucidated [44], these findings remain concerning, suggesting that a strategy of periodical immunization should be evaluated if the pandemic does not decrease its morbidity.

Italy, as well as several other countries, has established a different set of restriction measures according to vaccination status, and unvaccinated subjects have been denied access to a number of public or private facilities [45]. As a consequence, vaccinated subjects may have been exposed to a higher risk of SARS-CoV-2 transmission than unvaccinated individuals [21]. This clear bias may explain, at least in part, why the effectiveness of current vaccines against infection was modest at any dose.

In line with an ample amount of the literature, the number of COVID-19 hospitalizations was very low, with no deaths among the individuals younger than 30 years [14,18,21,41,46]. Given that the incidence of COVID-19 hospitalization was as low as 1 out of 2000 infections for unvaccinated individuals, and even though two vaccine doses were able to reduce the risk of hospitalization by 90%, the risk–benefit profile of multiple vaccine doses for this population should be carefully evaluated.

The overall number of serious adverse events following vaccination that was reported by the surveillance system of the Abruzzo region was very low, in agreement with the acceptable safety profile reported by all previous studies [47,48]. However, we had no access to individual-level data and, as correctly admitted by the operators of the system, the overall number of reports was extremely scarce, suggesting a low efficacy of the current, passive pharmacovigilance system for monitoring vaccine safety [49]. Furthermore, more detailed data are strongly needed to properly assess the harm–benefit profile of vaccination for specific clusters of the population [50].

The strengths of this study are the use of official, routinely collected electronic health databases from the entire, general population of an Italian region, followed for a maximum of 12 months. Importantly, individual-level data were available on several comorbidities that may influence the risk of COVID-19 hospitalization and death, and the analyses were accordingly adjusted for. Finally, although the follow-up was inevitably short, we had the chance to evaluate vaccine effectiveness during the first seven weeks of the Omicron wave, when more than 140,000 new infections were recorded.

This study also has limitations that must be considered when interpreting the results. Some of them have been mentioned: a relatively short follow-up after the booster dose and a higher contagion risk of the vaccinated vs. unvaccinated subjects, resulting from the less stringent restrictions, that may lead to an underestimation of vaccine effectiveness against infection. However, on the other side, under the regulations in force in Italy at the time of this study, unvaccinated individuals were more likely to undergo multiple testing (requested to attend social gatherings) [45], which may have enhanced their likelihood of a testing positive for COVID-19, potentially resulting in an overestimation of vaccine effectiveness for this outcome [51]. Finally, it may also be argued that the negative test case–control design is more appropriate for assessing real-world vaccine effectiveness in the context of a pandemic [52]; however, a good concordance with findings from cohort studies was observed by previous research, which directly compared the two methodologies [37]. Likewise, the observational study design was suboptimal for performing head-to-head comparisons of the diverse administered vaccines, all of which showed a similar effectiveness profile.

## 5. Conclusions

This large population-based cohort study confirmed that two or three doses of all current vaccines were able to confer a strong protection against COVID-19 hospitalization or death. During the Omicron wave, and six months after the last dose, protection decreased but remained high. The effectiveness against infection was modest at any dose, likely as a result of the differences in restriction policies by immunization status. Finally, the rate of reported serious adverse events was very low.

## Figures and Tables

**Figure 1 vaccines-10-00662-f001:**
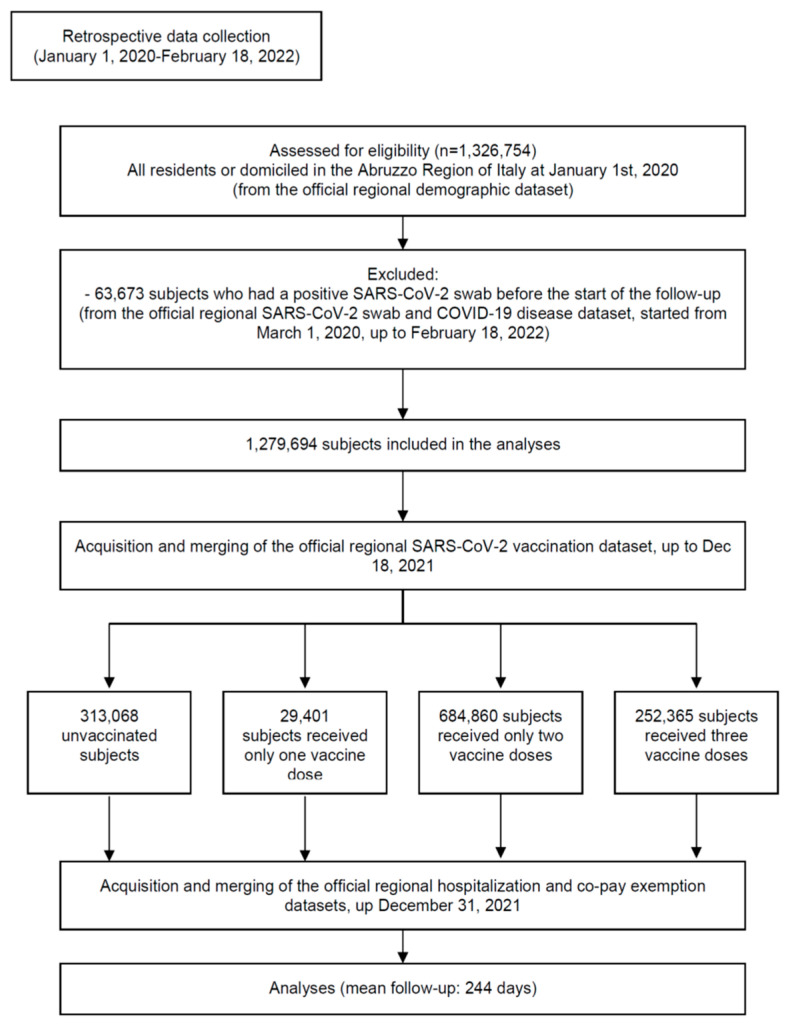
Study flowchart in line with the STROBE (Strengthening the Reporting of Observational Studies in Epidemiology) statement (http://www.strobestatement.org accessed on 16 April 2022).

**Table 1 vaccines-10-00662-t001:** Main characteristics of the sample, overall and by vaccine status.

	Unvaccinated	1 Dose ^A^	2 Doses ^B^	3 Doses ^C^	Total Sample
	(*n* = 313,068)	(*n* = 29,401)	(*n* = 684,860)	(*n* = 252,365)	(*n* = 1,279,694)
Gender					
Females	23.9	2.3	52.8	21.0	50.9 (651,752)
Males	25.1	2.3	54.2	18.4	49.1 (627,942)
Mean age in years (SD)	35.3 (26.6)	41.6 (21.6)	47.6 (19.7)	64.3 (16.9)	47.8 (23.3)
*Age category in years*					
0–29	45.6	3.3	48.4	2.7	25.3 (323,613)
30–59	19.2	2.4	61.8	16.7	42.0 (537,148)
60 or more	15.0	1.4	46.8	36.8	32.7 (418,933)
*Risk factors and comorbidities* ^D^					
No hypertension	25.8	2.4	55.3	16.4	87.4 (1,117,986)
Hypertension	14.9	1.4	41.1	42.6	12.6 (161,708)
No diabetes	24.9	2.3	54.4	18.4	94.7 (1,212,278)
Diabetes	16.8	1.4	38.4	43.3	5.3 (67,416)
No CVD	24.8	2.3	54.7	18.2	93.6 (1,198,232)
CVD	19.7	1.5	36.7	42.0	6.4 (81,462)
No COPD	24.5	2.3	53.8	19.4	97.5 (1,247,276)
COPD	25.1	2.0	42.2	30.7	2.5 (32,418)
No kidney disease	24.4	2.3	53.9	19.5	98.5 (1,260,117)
Kidney disease	30.3	1.6	31.7	36.4	1.5 (19,577)
No cancer	24.7	2.3	54.3	18.7	95.4 (1,220,868)
Cancer	20.6	1.2	36.7	41.5	4.6 (58,826)
*Type of vaccine* ^E^					
BNT162b2	--	64.6	65.4	59.7	63.9 (617,751)
mRNA-1273	--	29.7	14.2	7.6	13.0 (125,551)
ChAdOx1 nCoV-19	--	5.7	16.1	0.1	11.5 (111,658)
JNJ-78436735	--	--	1.7	0.0	1.2 (11,510)
Mixed ^F^	--	--	2.6	32.6	10.4 (100,156)
Mean follow-up in days (SD) ^G^	367 (58)	130 (87)	188 (51)	71 (18)	244 (99)
SARS-CoV-2-positive swab before the second dose ^H^	0.3 (877)	--	0.4 (2758)	0.1 (236)	0.3 (3871)
SARS-CoV-2-positive swab before the third dose ^H^	4.7 (14,835)	--	--	0.7 (1745)	1.3 (16,580)

SD = Standard deviation. If not differently stated, the values in the tables are expressed as % (*n*), where *n* = number of participants. Unless differently stated, the row percentages are shown: the sum of the percentages of the unvaccinated and those receiving 1, 2 or 3 doses, is 100%. ^A^ Subjects who received only one dose of BNT162b2, mRNA-1273 or ChAdOx1 nCoV-19 vaccines between 2 January 2021 and 18 December 2021.^B^ Subjects who received only one dose of JNJ-78436735 vaccine or only two doses of BNT162b2, mRNA-1273 or ChAdOx1 nCoV-19 vaccines between 2 January 2021 and 18 December 2021.^C^ Subjects who received two doses of JNJ-78436735 vaccine or three doses of BNT162b2, mRNA-1273 or ChAdOx1 nCoV-19 vaccines between 2 January 2021 and18 December 2021. ^D^ Subjects with the selected comorbidities in the regional co-pay exemption database (Italian “Esenzioni Ticket” file), or in the regional COVID database, or with hospital admission in the last ten years (from the Italian SDO database of administrative discharge abstracts) with the following ICD-9-CM codes in any diagnosis field: 250.xx (diabetes); 401.xx–405.xx (hypertension); 410.xx–412.xx or 414.xx–415.xx or 428.xx or 433.xx–436.xx (major cardiovascular or cerebrovascular diseases-CVD); 491.xx–493.xx (chronic obstructive pulmonary disease–COPD); 580.xx–589.xx (kidney diseases); and 140.xx–172.xx or 174.xx–208.xx (cancers). ^E^ For this variable, column percentages are reported. ^F^ Subjects who received two or three different vaccines. ^G^ The end of follow-up was the date of the first positive swab or 18 February 2022 for all the subjects that did not have a positive swab. The start of follow-up varied across vaccine categories: (a) 14 days after the single vaccine dose, for the group “1 dose”; (b) 14 days after the second vaccine dose, for the group “2 doses”; (c) 14 days after the third vaccine dose, for the group “3 doses”; (d) 14 days after the first administration of the second vaccine dose (31 January 2021) for the group “unvaccinated”. ^H^ The number of subjects who had a positive swab within 14 days of the reported vaccine dose.

**Table 2 vaccines-10-00662-t002:** Main outcomes, overall, by vaccine status, vaccine type, and age category.

Variables	Total Sample	Unvaccinated	1 Dose ^A^	2 Doses ^B^	3 Doses ^C^
	(*n* = 1,279,694)	(*n* = 313,068)	(*n* = 29,401)	(*n* = 684,860)	(*n* = 252,365)
**Health status**					
**Whole period**					
All subjects	*n* = 1,275,823 ^D^	*n* = 312,191	*n* = 29,401	*n* = 682,102	*n* = 250,620
SARS-CoV-2-positive swab	13.4 (170,761)	19.1 (59,770)	13.8 (4053)	13.1 (89,236)	6.5 (16,193)
COVID-19 hospitalization	0.15 (1910)	0.41 (1270)	0.22 (66)	0.06 (390)	0.06 (162)
COVID-19-related death	0.08 (1050)	0.28 (872)	0.13 (39)	0.02 (105)	0.01 (31)
Infected subjects only	*n* = 170,761 ^D^	*n* = 59,770	*n* = 4053	*n* = 89,236	*n* = 16,193
COVID-19 among the infected	1.12 (1910)	2.12 (1270)	1.63 (66)	0.44 (390)	1.00 (162)
Death among the infected	0.61 (1050)	1.46 (872)	0.96 (39)	0.12 (105)	0.19 (31)
**Pre-Omicron Waves** ^E^					
All subjects	*n* = 1,275,823 ^D^	*n* = 312,191	*n* = 29,401	*n* = 682,102	*n* = 250,620
SARS-CoV-2-positive swab	2.4 (30,726)	5.9 (18,489)	3.4 (1013)	1.4 (9449)	0.1 (266)
COVID-19 hospitalization	0.09 (1152)	0.31 (975)	0.15 (44)	0.02 (107)	0.00 (4)
COVID-19-related death	0.07 (933)	0.27 (831)	0.12 (35)	0.01 (63)	0.00 (1)
Infected subjects only	*n* = 30,726 ^D^	*n* = 18,489	*n* = 1013	*n* = 9449	*n* = 266
COVID-19 among the infected	3.75 (1152)	5.27 (975)	4.34 (44)	1.13 (107)	1.50 (4)
Death among the infected	3.04 (933)	4.49 (831)	3.46 (35)	0.67 (63)	0.38 (1)
**Omicron Wave** ^F^					
All subjects	*n* = 1,245,097 ^D^	*n* = 293,702	*n* = 28,388	*n* = 672,653	*n* = 250,354
SARS-CoV-2-positive swab	11.2 (140,035)	14.1 (41,281)	10.7 (3040)	11.9 (79,787)	6.4 (15,927)
COVID-19 hospitalization	0.06 (758)	0.10 (295)	0.08 (22)	0.04 (283)	0.06 (158)
COVID-19-related death	0.01 (117)	0.01 (41)	0.01 (4)	0.01 (42)	0.01 (30)
Infected subjects only	*n* = 140,035 ^D^	*n* = 41,281	*n* = 3040	*n* = 79,787	*n* = 15,927
COVID-19 among the infected	0.54 (758)	0.71 (295)	0.72 (22)	0.35 (283)	0.99 (158)
Death among the infected	0.08 (117)	0.10 (41)	0.13 (4)	0.05 (42)	0.19 (30)
**Age category**					
0–29 years	*n* = 322,643 ^D^	*n* = 146,948	*n* = 10,567	*n* = 156,352	*n* = 8658
SARS-CoV-2-positive swab	21.8 (70,345)	26.2 (38,494)	16.6 (1755)	18.5 (28,895)	12.9 (1113)
COVID-19 hospitalization	0.01 (24)	0.01 (19)	0.00 (0)	0.00 (4)	0.01 (1)
COVID-19-related death	0.00 (0)	0.00 (0)	0.00 (0)	0.00 (0)	0.00 (0)
30–59 years	*n* = 535,525 ^D^	*n* = 102,594	*n* = 12,873	*n* = 330,649	*n* = 88,639
SARS-CoV-2-positive swab	13.8 (73,852)	15.8 (16,177)	14.1 (1820)	14.1 (46,473)	9.7 (8612)
COVID-19 hospitalization	0.05 (253)	0.18 (189)	0.05 (6)	0.01 (46)	0.01 (8)
COVID-19-related death	0.01 (78)	0.07 (74)	0.00 (0)	0.00 (2)	0.00 (2)
60+ years	*n* = 417,655 ^D^	*n* = 62,649	*n* = 5961	*n* = 195,101	*n* = 153,323
SARS-CoV-2-positive swab	6.3 (26,534)	8.1 (5099)	8.0 (478)	7.1 (13,868)	4.2 (6468)
COVID-19 hospitalization	0.39 (1633)	1.70 (1062)	1.01 (60)	0.17 (340)	0.10 (153)
COVID-19-related death	0.23 (972)	1.27 (798)	0.65 (39)	0.05 (103)	0.02 (29)
**Vaccine type**					
BNT162b2	*n* = 615,841 ^D^	--	*n* = 18,994	*n* = 446,235	*n* = 149,708
SARS-CoV-2-positive swab	12.2 (74,886)	--	14.3 (2709)	13.8 (61,751)	6.36 (9522)
COVID-19 hospitalization	0.08 (477)	--	0.29 (56)	0.06 (279)	0.09 (130)
COVID-19-related death	0.02 (148)	--	0.18 (34)	0.02 (88)	0.02 (25)
mRNA-1273	*n* = 125,162 ^D^	--	*n* = 8727	*n* = 97,237	*n* = 19,098
SARS-CoV-2-positive swab	11.5 (14,458)	--	12.2 (1067)	12.7 (12,369)	4.8 (922)
COVID-19 hospitalization	0.05 (61)	--	0.10 (9)	0.04 (39)	0.07 (13)
COVID-19-related death	0.01 (16)	--	0.05 (4)	0.01 (9)	0.02 (3)
ChAdOx1 nCoV-19	*n* = 111,570 ^D^	--	*n* = 1680	*n* = 109,874	*n* = 16
SARS-CoV-2-positive swab	10.5 (11,679)	--	16.5 (277)	10.4 (11,401)	6.25 (1)
COVID-19 hospitalization	0.06 (65)	--	0.06 (1)	0.06 (64)	0.00 (0)
COVID-19-related death	0.01 (8)	--	0.06 (1)	0.01 (7)	0.00 (0)
JNJ-78436735	*n* = 11,510 ^D^	--	--	*n* = 11,508	*n* = 2
SARS-CoV-2-positive swab	13.1 (1508)	--	--	13.1 (1508)	0.00 (0)
COVID-19 hospitalization	0.06 (7)	--	--	0.06 (7)	0.00 (0)
COVID-19-related death	0.00 (0)	--	--	0.00 (0)	0.00 (0)
Mixed ^G^	*n* = 99,549 ^D^	--	--	*n* = 17,248	*n* = 81,796
SARS-CoV-2-positive swab	8.5 (8460)	--	--	12.8 (2207)	7.0 (5748)
COVID-19 hospitalization	0.02 (22)	--	--	0.01 (1)	0.02 (19)
COVID-19-related death	0.01 (6)	--	--	0.01 (1)	0.00 (3)
**Follow-up duration** ^H^					
Subjects infected within 6 months from the last dose	*n* = 102,216 ^D^	*n* = 8973	*n* = 3654	*n* = 71,888	*n* = 16,192
COVID-19 among the infected	1.24 (1267)	9.48 (851)	1.51 (55)	0.25 (177)	1.00 (162)
Death among the infected	0.86 (883)	8.47 (760)	0.96 (35)	0.08 (54)	0.19 (31)
Subjects infected after 6 months from the last dose	*n* = 68,545 ^D^	*n* = 50,797	*n* = 399	*n* = 17,348	*n* = 1
COVID-19 among the infected	0.94 (643)	0.82 (419)	2.76 (11)	1.23 (213)	0.00 (0)
Death among the infected	0.24 (167)	0.22 (112)	1.00 (4)	0.29 (51)	0.00 (0)

If not differently stated, the values in the tables are expressed as % (*n*), where *n* = number of participants. COVID-19 hospitalization = virologically confirmed COVID-19 disease, diagnosed by a specialist physician and requiring hospital admission. ^A^ Subjects who received only one dose of BNT162b2, mRNA-1273 or ChAdOx1 nCoV-19 vaccines between 2 January 2021 and 18 December 2021. ^B^ Subjects who received only one dose of JNJ-78436735 vaccine or only two doses of BNT162b2, mRNA-1273 or ChAdOx1 nCoV-19 vaccines between 2 January 2021 and 18 December 2021. ^C^ Subjects who received two doses of JNJ-78436735 vaccine, or three doses of BNT162b2, mRNA-1273 or ChAdOx1 nCoV-19 vaccines, between 2 January 2021 and 18 December 2021. ^D^ The number of the total does not equal the sum of the numbers of subjects in each category because both the start and the end of follow-up varied across the categories. As an example, as reported in Table 1 footnote “^G^”, for the analyses comparing the group “1 dose” versus “unvaccinated”, the follow-up started 14 days after the day of the single vaccine dose or, for the unvaccinated, on 16 January 2021 (14 days after the start of the immunization campaign). For the category “2 doses”, the follow-up started 14 days after the second dose or, for the unvaccinated, on 31 January 2021 (14 days after the start of the administration of the second doses). Therefore, the subjects who tested positive for SARS-CoV-2 between 16th January and 31st January 2021 were included in the comparison “dose 1 versus no vaccination”; however, they were excluded from the comparison “dose 2 versus no vaccination” as they were infected before the start of the follow-up of the second dose. Given that the main comparison was “two doses vs. none”, the numbers in the categories “total” and “unvaccinated” are the numbers of the subjects that were included in the comparison “2 doses” and may not be equal to the sum of the three categories. ^E^ Includes only the outcomes that occurred from the start of the follow-up to 26 December 2021, when the proportion of Omicron variant in the available positive swabs was lower than 50%. The average follow-up of the “3 doses” group was very short (19 days) during the pre-Omicron period. ^F^ Includes only the outcomes that occurred from 27 December 2021 to 18 February 2022, when the proportion of Omicron variant in the available positive swabs was higher than 50%. The subjects who were infected before the predominance of Omicron were excluded from the sample. ^G^ Subjects who received two or three different vaccines. ^H^ Two separate analyses were conducted: the first only included events that occurred 182 days (6 months) from the last vaccine dose (start of follow-up); the second only included the events that occurred >6 months after the last dose of vaccine. The subjects with follow-up shorter than 6 months, as well as the subjects that had a positive swab within 6 months of follow-up, were excluded from the analysis of the events occurring after six months of follow-up.

**Table 3 vaccines-10-00662-t003:** Multivariable analysis ^ψ^ of the effectiveness of COVID-19 vaccines.

Variables	SARS-CoV-2	COVID-19 Hospitalization ^A^	COVID-19-Related Death
	OR (95% CI)	OR (95% CI)	OR (95% CI)
**Health status**			
**Whole period**			
Vaccine doses			
All subjects			
Unvaccinated	1 (Ref. cat.)	1 (Ref. cat.)	1 (Ref. cat.)
1 dose	0.71 (0.68–0.73) *	0.53 (0.41–0.67) *	0.49 (0.35–0.67) *
2 doses	0.75 (0.74–0.76) *	0.12 (0.10–0.13) *	0.06 (0.05–0.07) *
3 doses	0.74 (0.73–0.76) *	0.21 (0.18–0.26) *	0.24 (0.16–0.37) *
Infected subjects only			
Unvaccinated	--	1 (Ref. cat.)	1 (Ref. cat.)
1 dose	--	0.44 (0.33–0.58) *	0.33 (0.23–0.47) *
2 doses	--	0.10 (0.09–0.11) *	0.05 (0.04–0.06) *
3 doses	--	0.18 (0.15–0.23) *	0.15 (0.10–0.24) *
**Omicron wave** ^B^			
All subjects			
Unvaccinated	1 (Ref. cat.)	1 (Ref. cat.)	1 (Ref. cat.)
2 doses	0.98 (0.97–1.00)	0.34 (0.29–0.40) *	0.68 (0.43–1.09)
3 doses	0.81 (0.80–0.83) *	0.25 (0.20–0.30) *	0.42 (0.26–0.68) *
Infected subjects only			
Unvaccinated	1 (Ref. cat.)	1 (Ref. cat.)	1 (Ref. cat.)
2 doses	--	0.21 (0.18–0.25) *	0.31 (0.20–0.49) *
3 doses	--	0.19 (0.16–0.24) *	0.23 (0.14–0.37) *
**Vaccine type** (2 doses vs. none)			
Unvaccinated	1 (Ref. cat.)	1 (Ref. cat.)	1 (Ref. cat.)
BNT162b2	0.76 (0.75–0.77) *	0.14 (0.12–0.16) *	0.08 (0.06–0.10) *
mRNA-1273	0.68 (0.66–0.69) *	0.10 (0.07–0.14) *	0.04 (0.02–0.08) *
ChAdOx1 nCoV-19	0.96 (0.94–0.99) **	0.07 (0.05–0.08) *	0.02 (0.01–0.04) *
JNJ-78436735	0.88 (0.83–0.93) *	0.13 (0.06–0.27) *	0.00 (NE)
Mixed ^C^	0.76 (0.72–0.79) *	0.02 (0.00–0.16) *	0.04 (0.01–0.28) *
**Age category**, years (2 doses vs. none)			
60 or more	0.79 (0.76–0.82) *	0.14 (0.12–0.16) *	0.06 (0.05–0.07) *
30–59	0.90 (0.88–0.91) *	0.07 (0.05–0.10) *	0.01 (0.00–0.03) *
0–29	0.85 (0.83–0.87) *	0.10 (0.03–0.30) *	NE

OR = odds ratio; CI = confidence interval; Ref. cat. = reference category. NE = not estimable (0 cases in one or both of the groups under comparison). * *p* < 0.001; ** *p* < 0.05. ^ψ^ Logistic regression models adjusted for age, gender, hypertension, diabetes, major cardiovascular diseases, chronic obstructive pulmonary diseases, kidney diseases, and cancer. ^A^ Virologically confirmed COVID-19 disease, diagnosed by a specialist physician and requiring hospital admission. ^B^ Includes only the outcomes that occurred from 27 December 2021 to 18 February 2022, when the proportion of Omicron variant in the available positive swabs was higher than 50%. ^C^ Subjects who received two or three different vaccines.

**Table 4 vaccines-10-00662-t004:** Multivariable analysis ^ψ^ predicting vaccine effectiveness to prevent COVID-19 hospitalization or death among the infected subjects, according to the duration of follow-up.

Variables	COVID-19 Hospitalization ^A^	COVID-19-Related Death
**Follow-up duration** ^B^	OR (95% CI)	OR (95% CI)
≤6 months of follow-up		
Unvaccinated	1 (Ref. cat.)	1 (Ref. cat.)
2 doses	0.03 (0.02–0.03) *	0.01 (0.01–0.02) *
3 doses	0.18 (0.15–0.23) *	0.15 (0.10–0.24) *
>6 months of follow-up		
Unvaccinated	1 (Ref. cat.)	1 (Ref. cat.)
2 doses	0.31 (0.26–0.37) *	0.25 (0.17–0.35) *
3 doses	NE	NE

OR = odds ratio; CI = confidence interval; Ref. cat. = reference category. NE = not estimable (0 cases in one or both of the groups under comparison). * *p* < 0.001. ^ψ^ Logistic regression models adjusted for age, gender, hypertension, diabetes, major cardiovascular diseases, chronic obstructive pulmonary diseases, kidney diseases, and cancer. ^A^ Virologically confirmed COVID-19 disease, diagnosed by a specialist physician and requiring hospital admission. ^B^ Two separate analyses were conducted: the first only included the events that occurred 182 days (6 months) from the start of follow-up, which was censored at 182 days; the second only included the events that occurred >6 months after the start of follow-up. The subjects with follow-up shorter than 6 months, as well as subjects that had a positive swab within 6 months of follow-up, were excluded from the analysis of the events occurring after the 6 months of follow-up.

## Data Availability

The data presented in this study are available upon reasonable request from the corresponding author.

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
