# Peer review of "Effectiveness of COVID-19 Vaccines in the General Population of an Italian Region before and during the Omicron Wave"

_vaccines, 2022, doi:10.3390/vaccines10050662_

Round 1
Reviewer 1 Report
The report by Martellucci et al. is a well-written, important contribution to the vaccination outcome literature on COVID-19, and supplements other reports. It has important strengths, such as the large numbers, the reported co-morbidities, and age distribution.
I have only a few minor comments. The use of the terms severe, disease, hospitalization and COVID-disease are used somewhat loosely and interchangeably throughout the paper (see Table 2 where this condition is then referred to as only “COVID-19 among the infected”). I would prefer that one term be used throughout the paper, and as this is defined by hospitalization, that would be my preferred term.
Second, we need to know why they broke down the cohorts by the specific age ranges (0-29, 30-59, and 60+). Were other ranges evaluated, or was this due to how data is analyzed in Italy? Also the time form vaccination to outcomes is broken down to 182 days. Were other intervals assessed (for example 90 days?), and why not evaluate this as a continuous parameter?
Author Response
I-1. The Referee wrote "The report by Martellucci et al. is a well-written, important contribution to the vaccination outcome literature on COVID-19, and supplements other reports. It has important strengths, such as the large numbers, the reported co-morbidities, and age distribution.
I have only a few minor comments. The use of the terms severe, disease, hospitalization and COVID-disease are used somewhat loosely and interchangeably throughout the paper (see Table 2 where this condition is then referred to as only “COVID-19 among the infected”). I would prefer that one term be used throughout the paper, and as this is defined by hospitalization, that would be my preferred term".
We agree and accordingly replaced all terms with COVID-19 hospitalization throughout the manuscript and tables.
I-2. The Referee wrote "Second, we need to know why they broke down the cohorts by the specific age ranges (0-29, 30-59, and 60+). Were other ranges evaluated, or was this due to how data is analyzed in Italy? Also the time from vaccination to outcomes is broken down to 182 days. Were other intervals assessed (for example 90 days?), and why not evaluate this as a continuous parameter?"
We absolutely agree that the choice of both the age ranges and the time from vaccination cutoff should have been explained, and we accordingly revised the paragraph of the Methods section as follows: "Each of these comparisons were stratified by age-class (0-29y, 30-59y, 60+y), vaccine type, and time since the last vaccine dose (≤ or >182 days - six months - the duration of the "green pass" set by the Italian Government for the subjects who recovered after a SARS-CoV-2 infection) (31). The age ranges were identified to be consistent with the periodical reports of the Italian Institute of Health (32), and with the recommendations of the Italian Government (33), which identified as priority targets for vaccinations the subjects aged ≥60 years"
Reviewer 2 Report
In this manuscript, Martellucci et al. conducted a large population-based cohort study to assess the real-world effectiveness of various SARS-CoV-2 vaccines (including both novel mRNA-based vaccines by Pfizer and Moderna) to protect against infection and severe/lethal COVID-19 following immunization and during the height of the Omicron variant transmission compared to the unvaccinated population. The authors found that although protection decreased over time, the effectiveness of vaccination in preventing severe/lethal COVID-19 remained high. This study was quite interesting and provides needed information on the longer-term efficacy of the major COVID-19 vaccines. Some specific comments are given below:
(1) In the Abstract (line 27), change "this findings" to "these findings".
(2) In the Introduction (line 44), remove the superscript 22.
(3) A schematic diagram or table illustrating the dose groups vs. comparison groups, the start of follow-up, and the study outcomes would help to clarify the authors' cohort design that is described in the Methods section. The textual description was difficult to follow as written.
(4) Please remove the hyphens in Tables 1, 3, and 4. They serve no purpose.
Author Response
II-1. The Referee wrote "In this manuscript, Martellucci et al. conducted a large population-based cohort study to assess the real-world effectiveness of various SARS-CoV-2 vaccines (including both novel mRNA-based vaccines by Pfizer and Moderna) to protect against infection and severe/lethal COVID-19 following immunization and during the height of the Omicron variant transmission compared to the unvaccinated population. The authors found that although protection decreased over time, the effectiveness of vaccination in preventing severe/lethal COVID-19 remained high. This study was quite interesting and provides needed information on the longer-term efficacy of the major COVID-19 vaccines. Some specific comments are given below:
(1) In the Abstract (line 27), change "this findings" to "these findings".
(2) In the Introduction (line 44), remove the superscript 22.
(3) Please remove the hyphens in Tables 1, 3, and 4. They serve no purpose".
We agree and accordingly revised the manuscript as requested.
II-2. The Referee wrote "A schematic diagram or table illustrating the dose groups vs. comparison groups, the start of follow-up, and the study outcomes would help to clarify the authors' cohort design that is described in the Methods section. The textual description was difficult to follow as written".
We agree and accordingly added a online supplemental table (Table S1), reporting the details of the study groups, follow-up, and outcomes. We thank the Referee for the suggestion.
Reviewer 3 Report
Dear Authors,
thank you for your paper of current interest.
It gives a very good point of view about effectiveness of these new COVID-19 vaccines and results could be a starting point for further evaluations which could be useful for the decision-making about the management of health services and implications during pandemic.
However, it needs some major revisions to be clearer and adequately structured for publication. Please find p-2-p comments in the file attached.

Author Response
III-1. The Referee wrote "Dear Authors, thank you for your paper of current interest. It gives a very good point of view about effectiveness of these new COVID-19 vaccines and results could be a starting point for further evaluations which could be useful for the decision-making about the management of health services and implications during pandemic. However, it needs some major revisions to be clearer and adequately structured for publication.
Lines 124-127: the reasons behind the choice of the age classes should be explicated; the same should be done for the break at 182 days. I suppose it corresponds to the break at 6 months described in Table 2 and in Table 4. If this is the case, the same nomenclature should be used throughout the paper".
We absolutely agree that the choice of both the age ranges and the time from vaccination cutoff should have been explained, and we accordingly revised the first paragraph of the Data analysis as follows: "Each of these comparisons were stratified by age-class (0-29y, 30-59y, 60+y), vaccine type, and time since the last vaccine dose (≤ or >182 days - six months - the duration of the "green pass" set by the Italian Government for the subjects who recovered after a SARS-CoV-2 infection) (31). The age ranges were identified to be consistent with the periodical reports of the Italian Institute of Health (32), and with the recommendations of the Italian Government (33), which identified as priority targets for vaccinations the subjects aged ≥60 years"
We also agree that the same nomenclature should have been adopted, and we accordingly used "six months" throughout the manuscript.
III-2. The Referee wrote "Lines 132-134: is there a valid methodological reference supporting this choice?"
We agree and accordingly added the following reference in support: Pouwels KB, Pritchard E, Matthews PC, Stoesser N, Eyre DW, Vihta KD, et al. Effect of Delta variant on viral burden and vaccine effectiveness against new SARS-CoV-2 infections in the UK. Nature Medicine 2021;27(12):2127-35.
III-3. The Referee wrote "Lines 134-141: all results from these analyses should be provided as supplementary material; in particular, it would be very interesting to have public access to the results of multivariable logistic regressions".
We absolutely agree and accordingly added an online supplemental Table (S2), in which the complete results of the multivariable logistic regressions for each outcome are reported.
III-4. The Referee wrote "Lines 18, 230, 256, 265, 285: the use of the word “multivariable” instead of “multivariate” should be preferred".
We agree and accordingly replaced "multivariate" with "multivariable" throughout the manuscript.
III-5. The Referee wrote "Figure 1 (line 159): this flow chart is useful but could be improved by adhering to STROBE reporting standard for cohort studies and making it more schematic".
We agree and accordingly revised entirely the study flowchart according to STROBE reporting standards.
III-6. The Referee wrote "Table 1 (line 161): as well described in the caption, the column “Total sample” must be read vertically”, while the other four columns must be read horizontally together. This creates confusion. The authors should find a way to make it easier to read: maybe coloring the total sample column differently and/or moving it at the right extremity of the table".
We agree and accordingly moved the total sample column at the right extremity of the table. We thank the Referee for the suggestion.
III-7. The Referee wrote "Lines 188-191: the authors refer to data regarding pre-Omicron waves. Even if I were able to obtain them by subtraction from the available data, they should be made explicit in Table 2".
We agree and accordingly added the raw data regarding pre-Omicron waves in Table 2.
III-8. The Referee wrote "Table 2 (198): this table is not clear and must be improved. I would prefer “Health status” instead of “Infection status”. Furthermore, I would organize the presentation of data within “Health status” in three sub-levels, as follows:
|
| |
Total sample |
Unvaccinated |
1 Dose |
2 Dose |
3 Dose |
|
|Health status |
|
|
|
|
|
|
|Whole period |
|
|
|
|
|
|
|All subjects |
|
|
|
|
|
|
|… |
|
|
|
|
|
|
|Only infected subjects |
|
|
|
|
|
|
|… |
|
|
|
|
|
|
|Pre-Omicron waves |
|
|
|
|
|
|
|All subjects |
|
|
|
|
|
|
|… |
|
|
|
|
|
|
|Only infected subjects |
|
|
|
|
|
|
|… |
|
|
|
|
|
|
|Omicron wave |
|
|
|
|
|
|
|All subjects |
|
|
|
|
|
|
|… |
|
|
|
|
|
|
|Only infected subjects |
|
|
|
|
|
|
|… |
|
|
|
|
|
|
|Age class … |
|
|
|
|
|
Or I would present them as interaction variables:
|
| |
Total sample |
Unvaccinated |
1 Dose |
2 Dose |
3 Dose |
|
|Health status, whole period |
|
|
|
|
|
|
|All subjects |
|
|
|
|
|
|
|… |
|
|
|
|
|
|
|Only infected subjects |
|
|
|
|
|
|
|… |
|
|
|
|
|
|
|Health status, pre-Omicron waves |
|
|
|
|
|
|
|All subjects |
|
|
|
|
|
|
|… |
|
|
|
|
|
|
|Only infected subjects |
|
|
|
|
|
|
|… |
|
|
|
|
|
|
|Health status, Omicron wave |
|
|
|
|
|
|
|All subjects |
|
|
|
|
|
|
|… |
|
|
|
|
|
|
|Only infected subjects |
|
|
|
|
|
|
|… |
|
|
|
|
|
|
|Age class … |
|
|
|
|
|
|
|
|
|
|
|
|
We agree and thank the Referee for the suggestion. We accordingly replaced "Infection status" with "Health status" and re-organized the presentation of data within “Health status” in three sub-levels, as suggested.
III-9. The Referee wrote "Line 251: since the authors have calculated p values, I would recommend including them in Table 3 and Table 4, preferably in the form of differentiated starring according to significance levels".
We agree and accordingly added differentiated stars according to the significance level in both Table 3 and Table 4.
III-10. The Referee wrote "Lines 252-255: I would decouple the comments regarding the Omicron wave and the duration of follow-up or, at least, I would specify “decreased to approximately 66% and 69%, respectively”".
We agree and accordingly rephrased the sentence as suggested. We thank the Referee for the correction.
III-11. The Referee wrote "Table 3 (line 256): this table should be improved; I would recommend similar changes as for Table 2.
Another possibility would be to make two different sub-tables for:
- Whole period:
- Vaccine doses
- All subjects
- Only infected subjects
- Vaccine type
- Age-class
- Omicron wave:
- Vaccine doses
- All subjects
- Only infected subjects
- ....
We agree and accordingly re-organized Table 3 presentation of data as suggested.
III-12. The Referee wrote "It would be interesting to see the results for the same analysis for the isolated pre-Omicron waves, if available".
We agree and thank the Referee for the suggestion. Accordingly, the results of the main analyses were replicated for the isolated pre-Omicron waves, and the results were reported in a new online supplemental table (S3). Please acknowledge that we also mentioned the only different finding in the text, within the Results section, as follows: "On the contrary, during the pre-Omicron waves the risk of infection was significantly lower among the subjects who received any dose of vaccine (online Table S3)".
III-13. The Referee wrote "Table 4 (line 265-266): I do not understand why vaccine effectiveness in preventing infection was not evaluated with regard to follow-up duration. Furthermore, I believe the caption could be clearer: “according to the duration of follow-up”".
We agree and we are sorry for the oversight, we forgot to mention that the outcomes in Table 4 were computed among the infected subjects only, in order to exclude the potential distortion due to variation of the infection rates across the follow-up. Please acknowledge that we corrected the Table caption as follows: "Multivariable analysis ψ predicting vaccine effectiveness to prevent COVID-19 hospitalization or death among the infected subjects, according to the duration of follow-up".
III-14. The Referee wrote "Discussion and Conclusion: there is no Conclusion; the authors might consider adapting part of the Discussion as a Conclusion".
We agree and accordingly added a separate "Conclusion" paragraph.
III-15. The Referee wrote "Institutional Review Board Statement, Informed Consent Statement, Data Availability Statement (lines 395-401): more information should be given, at lines 395-401 or within the text, concerning measures undertaken to preserve patients’ privacy, with specific reference to the GDPR regulation. The retrospective and pseudo-anonymized nature of the data is not sufficient, per se, to waive patient consent for research purposes. Furthermore, the owner of the data processing is never specified, nor is specified at which level the pseudo-anonymization has been realized. Similarly, I wonder at what capacity the corresponding author might make raw data available upon request, as declared in the Data Availability Statement, without violating GDPR provisions, unless he is legitimately authorized to do so by the owner of the data processing".
We entirely agree that these important points should have been clarified, and tried to clarify them adding the following sentences in the Institutional Review Board Statement, Informed Consent Statement, Data Availability Statement: "According to the European Union General Data Protection (GDPR) regulation, all datasets were pseudo-anonymized by the Regional Offices before the access by the authors, using a unique identification code for each subject in each dataset. All information concerning the addresses, phone, email, date of birth, vaccination center, hospital site, swab lab, and municipality of all subjects were not provided to the authors, and the encrypted identification code could not be reversed by the authors or by the Regional Offices (the encryption was made in two steps, assigning random codes for each fiscal code in the demographic database, and the intermediate codes have been deleted). RC was the owner of the data processing and has granted LM the permission to release anonymized raw data upon request".
III-16. The Referee wrote "References: References should be presented according to MDPI and Chicago Style guidelines. In particular, for every reference all authors should be listed: in their present form, with only the first author listed, they make it difficult for the reviewer to fully assess the presence of inappropriate or excessive self-citations".
We agree and accordingly formatted the references using the MDPI Style guidelines.
III-17. The Referee wrote "Reference 10 (cited at line 35) and 24 (cited at line 51) are duplicates and must be united".
We agree and accordingly corrected the error. Please accept our apologies for the oversight.
Reviewer 4 Report
In the manuscript, the authors have studied the effectiveness of different COVID-19 vaccines in the general population of Abruzzo, Italy before and during the Omicron wave. Below I offer several comments regarding the manuscript.
- For this study, I would recommend adding a Statistical analyses section in the material and methods.
- In addition, another section should describe the data collection procedure.
- Information on the comorbidity index for the unvaccinated and vaccinated (one dose, two, and three doses) populations would be useful.
- Furthermore, I would like to know whether any one of the subjects selected for this cohort study has evidence of previous infection (PCR-positive SARS-CoV-2)?
Author Response
IV-1. The Referee wrote "In the manuscript, the authors have studied the effectiveness of different COVID-19 vaccines in the general population of Abruzzo, Italy before and during the Omicron wave. Below I offer several comments regarding the manuscript.
For this study, I would recommend adding a Statistical analyses section in the material and methods. In addition, another section should describe the data collection procedure".
We agree and accordingly added two separate section, one on the Statistical Analyses and one on the Data Collection procedure.
IV-2. The Referee wrote "Information on the comorbidity index for the unvaccinated and vaccinated (one dose, two, and three doses) populations would be useful".
We absolutely agree that such information would be useful. However, please acknowledge that unfortunately we did not ask these data at the beginning of the study, and it would be impossible now to request and merge the necessary additional information because of the complete anonymization that we made of the sensitive data. Unfortunately, to comply with the privacy regulations, at this point we cannot make a linkage between the other collected databases and new ones, as the fiscal codes or other id information have been encrypted, and this encryption cannot be reversed. We are sorry for that.
IV-3. The Referee wrote "Furthermore, I would like to know whether any one of the subjects selected for this cohort study has evidence of previous infection (PCR-positive SARS-CoV-2)?"
Please acknowledge that we excluded from the sample all the subjects with evidence of a previous infection. We accordingly reported this important point as follows: "Here, we included all the subjects resident or domiciled in the Abruzzo Region of Italy on 1 January 2020, with no positive SARS-CoV-2 swab before the start of follow-up, in order to evaluate the effectiveness of SARS-CoV-2 vaccination (one, two or three doses versus none) against infection..."
Round 2
Reviewer 3 Report
Dear Authors,
thank you for your answers. From my viewpoint, the paper is now suitable for publication.
Reviewer 4 Report
The manuscript ' Effectiveness of COVID-19 vaccines in the general population of an Italian Region before and during the Omicron wave' can be accepted in its present form.